# Effectiveness of nutritional countermeasures in microgravity and its ground-based analogues to ameliorate musculoskeletal and cardiopulmonary deconditioning–A Systematic Review

**Peter H. Sandal**[1]*, **David Kim**[1,2], **Leonie Fiebig**[1,3], **Andrew Winnard**[4], **Nick Caplan**[4], **David A. Green**[1,5,6], **Tobias Weber**[1,6]*

**1** Space Medicine Team, European Astronaut Centre, European Space Agency, Köln, Germany, **2** Faculty of Medicine, University of British Columbia, Vancouver, Canada, **3** Institute of Biomechanics und Orthopaedics, German Sport University, Cologne, Germany, **4** Faculty of Health and Life Sciences, Northumbria University, Newcastle upon Tyne, United Kingdom, **5** Centre of Human & Applied Physiological Sciences, King's College London, London, United Kingdom, **6** KBR GmbH, Köln, Germany

* Tobias.weber@esa.int (TW); Peterhsandal@gmail.com (PS)

## Abstract

A systematic review was performed to evaluate the effectiveness of nutrition as a standalone countermeasure to ameliorate the physiological adaptations of the musculoskeletal and cardiopulmonary systems associated with prolonged exposure to microgravity. A search strategy was developed to find all astronaut or human space flight bed rest simulation studies that compared individual nutritional countermeasures with non-intervention control groups. This systematic review followed the guidelines of the Cochrane Handbook for Systematic Reviews and tools created by the Aerospace Medicine Systematic Review Group for data extraction, quality assessment of studies and effect size. To ensure adequate reporting this systematic review followed the guidelines of the Preferred Reporting Items for Systematic Review and Meta-Analyses. A structured search was performed to screen for relevant articles. The initial search yielded 4031 studies of which 10 studies were eligible for final inclusion. Overall, the effect of nutritional countermeasure interventions on the investigated outcomes revealed that only one outcome was in favor of the intervention group, whereas six outcomes were in favor of the control group, and 43 outcomes showed no meaningful effect of nutritional countermeasure interventions at all. The main findings of this study were: (1) the heterogeneity of reported outcomes across studies, (2) the inconsistency of the methodology of the included studies (3) an absence of meaningful effects of standalone nutritional countermeasure interventions on musculoskeletal and cardiovascular outcomes, with a tendency towards detrimental effects on specific muscle outcomes associated with power in the lower extremities. This systematic review highlights the limited amount of studies investigating the effect of nutrition as a standalone countermeasure on operationally relevant outcome parameters. Therefore, based on the data available from the included studies in this systematic review, it cannot be expected that nutrition alone will be

**Data Availability Statement:** All relevant data are within the manuscript and its Supporting Information files.

**Funding:** The project was funded by the Space Medicine Team of the European Space Agency and KBR GmbH. The funder (KBR GmbH) provided support in the form of salaries for authors [DAG, TW] but did not have any additional role in the study design, data collection and analysis, decision to publish, or preparation of the manuscript. The specific roles of these authors are articulated in the 'author contributions' section.

**Competing interests:** The funder (KBR GmbH) provided support in the form of salaries for authors [DAG, TW] but did not have any additional role in the study design, data collection and analysis, decision to publish, or preparation of the manuscript. This does not alter our adherence to PLOS ONE policies on sharing data and materials.

effective in maintaining musculoskeletal and cardiopulmonary integrity during space flight and bed rest.

## Introduction

Prolonged exposure to microgravity (μg) leads to a range of adaptations affecting almost every physiological system in the human body [1]. Musculoskeletal and cardiopulmonary de-conditioning are of particular concern to spaceflight operations, since these systems are critical for astronaut health and wellbeing and potentially even safety and mission success. Consequently, space agencies employ in-flight exercise countermeasures as an integral part of long-duration international space station (ISS) (typically 6 months) space missions [2,3].

In space, muscles atrophy and lose both power and strength [4,5], particularly in the so-called 'antigravity muscles' such as the soleus [4,6,7] and the intrinsic back muscles [5,7,8]. As both an increase in protein breakdown and a decrease in anabolism can lead to a decrease in muscle mass [9], μg-induced loss of muscle mass is likely to be associated with space-related changes of whole-body protein turnover. In fact, studies indicate that during short-duration space flight whole body protein turnover increases, with a more pronounced increase in protein catabolism compared to protein synthesis leading to a net loss of protein [10–12]. In addition, nutritional deficiencies including a reduction in caloric intake compared to pre-flight, commonly reported inflight, might attenuate maintenance of protein synthesis [13], thereby exacerbating μg-induced muscle deconditioning [14]. This is supported by a long duration study from the Mir space station where it was observed that protein synthesis was directly correlated with energy intake [15].

Bone loss during spaceflight is estimated to be approximately 1% per month, primarily affecting the weight-bearing bones, although there appears significant inter-individual variation between crewmembers [16]. Interestingly, during space flight where the gravitational forces acting on bones are absent, increased bone resorption is not offset by augmentation of bone formation [9], which is either decreased or unchanged [9]. The relationship between nutrition and bone is complex involving several nutrients including protein required for bone synthesis [9]. However, studies investigating the effect of high protein intake on bone health have been inconclusive [9,17] and it has been speculated that excess protein intake may augment bone loss due to sulphur-containing amino acids reacting with bicarbonate stored in bone, thereby depleting its reservoirs [9,18].

When exposed to μg the hydrostatic gradient associated with upright posture in Earth's gravity is lost, and blood is redistributed towards the head and neck [19]. As a result, on return to Earth astronauts can experience orthostatic intolerance, the incidence rate of which increases with mission duration [20–22]. Orthostatic intolerance is of particular concern as it constitutes a serious hazard for future planetary exploration missions where astronauts may spend a prolonged period in μg before entering a gravitational field of a celestial body (e.g. Mars) where they have to immediately perform mission-critical tasks without medical support [23,24]. Interestingly, deficient energy intake has been correlated with cardiovascular deconditioning and a decrease in plasma volume during both bed rest–a ground-based analogue of space flight [25], and space flight [9] and has been shown to negatively affect orthostatic tolerance independent from bed rest [25]. Furthermore, poor nutrition during space flight might also adversely affect cardiopulmonary outcomes by precipitating lean tissue loss leading to a reduction in metabolically active tissue, thereby directly reducing (peripheral) aerobic capacity

[26]. Moreover, insufficient protein intake might augment a decrease in muscle mass negatively impacting exercise performance, and thus (indirectly) aerobic capacity [26]. However, it has been suggested that cardiac mass and functional decrements associated with bed rest might be ameliorated by nutritional countermeasures [26].

Despite the time, effort and resources consumed by current ISS exercise countermeasures they are unable to entirely ameliorate the undesirable effects of μg-exposure upon the musculoskeletal [27] and cardiopulmonary systems [28]. Furthermore, future space vehicles designed for deep spaceflight will have limited habitable volume including that for exercise countermeasures [29]. Thus, in order to support crew health and wellbeing, safety, and to increase the likelihood of mission success, it is therefore critical to evaluate approaches to optimize both current and novel countermeasures to maintain musculoskeletal and cardiopulmonary function inflight [30]. Typically crewmembers did not meet their recommended daily caloric intake which plays a role in subsequent loss of body and muscle mass [9]. This persistent inadequate intake is concerning as its effects will become increasingly severe as exploration missions lengthen. However, frequently the negative effects of dietary imbalance can be reversed by adequate nutrition. Interestingly when astronauts were able to meet their energy and vitamin-D intake requirements in addition to performing regular exercise it was reported that crewmembers returned to Earth with an unchanged body mass, increased lean tissue mass, reduced fat mass and maintained bone mineral density (BMD) [31]. However, it is currently unclear if any nutritional countermeasure alone can maintain integrity of the musculoskeletal and cardiopulmonary systems when humans are gravitationally unloaded. Thus, the aim of this systematic review was to evaluate the evidence relating to the effectiveness of any nutritional countermeasures, as a standalone intervention, to ameliorate musculoskeletal and cardiopulmonary deconditioning in gravitationally unloaded humans, either inflight or in long term ground-based analogues.

## Materials and methods

This systematic review is part of a series performed by the European Space Agency's (ESA) Space Medicine Team that seeks to evaluate the evidence relating to the effectiveness of active (e.g. resistive exercise), passive (e.g. centrifugation), and nutritional countermeasures for use in space flight, based on inflight data and ground-based analogues. This systematic review followed the guidelines of the Cochrane Handbook for Systematic Reviews and tools created by the Aerospace Medicine Systematic Review Group (AMSRG) [32,33] were used for data extraction, quality assessment of studies, and effect size calculations. Furthermore, this systematic review followed the guidelines of the Preferred Reporting Items for Systematic Review and Meta-Analyses (PRISMA) [34]. The review protocol is available as supporting information (see S1 File).

### Search strategy

An initial literature search was performed by Fiebig and co-workers [35] in November 2017 whom reported that resistive exercise may not always be sufficient in maintaining muscle strength and power during bed rest. This work has led to the seeding of a series of systematic reviews each with the objective of evaluating the evidence of a specific type of countermeasures, which in this case is nutrition on musculoskeletal and cardiopulmonary outcomes that are operationally relevant to space missions. The search was performed using keywords, grouped into three overarching categories 'microgravity', 'countermeasures' and 'operationally relevant outcome parameters' (as defined by ESA's Space Medicine Team) combined using Boolean logic in accordance with an overall search strategy (Table 1).

**Table 1. Search strategy.**

| Main category | Specific category | | Keywords in Boolean search format | Search number | Search mask |
|---|---|---|---|---|---|
| Microgravity | Synonyms | | "space analogue" OR "ground-based analogue" OR "terrestrial analogue" OR "space flight" OR space-flight OR spaceflight OR "Space mission" OR "space station" OR "micro gravity" OR micro-gravity OR microgravity OR spaceflight OR weightless* OR "orbital flight" OR "zero gravity" OR "space shuttle" | 1 | Abstract/ Title |
| | Methods & simulations | | "bed rest" OR bed rest OR "dry immersion" OR dry-immersion | 2 | Abstract/ Title |
| | | | #1 AND #2 | 3 | |
| | Population of interest | | Astronaut* OR astronaut [Mesh] OR cosmonaut* OR taikonaut* | 4 | Abstract/ Title |
| | | | #1 OR #3 OR #4 | 5 | |
| Countermeasures | Active countermeasures | | Countermeasure* OR exercis* OR exercise [Mesh] OR sport* OR "physical activity" OR "physically active" | 6 | All Fields |
| | Passive countermeasures | | Centrifug* OR suit* OR "lower body negative pressure" OR LBNP or "fluid loading" OR garment OR stimulation OR "artificial gravity" OR "axial loading" OR electromyostimulation OR "electrical muscle stimulation" OR EMS OR "neuromuscular electrical stimulation" OR NMES OR "whole body vibration" OR WBV | 7 | All Fields |
| | Nutritional countermeasures | | Diet, food, and nutrition [Mesh] OR nutrition* OR diet* OR food* OR supplement* OR protein* OR salt OR saline OR bi-phosphonate OR phosphonate OR nucleotide* OR vitamin* | 8 | All Fields |
| | | | #6 OR #7 OR #8 | 9 | |
| Operationally relevant outcome parameters | Cardiopulmonary & -vascular | Physical performance | "endurance" OR Vo2 OR Vo2max OR Vo2peak OR "maximal oxygen uptake" OR "peak oxygen uptake" OR "resting heart rate" OR "peak power" OR "maximal work load" OR "orthostatic tolerance" OR "orthostatic intolerance" OR "time until presyncope" OR "exercise tolerance" OR "central fatigue" OR "threshold" OR "onset of blood lactate accumulation" OR "OBLA" | 10 | All Fields |
| | Musculoskeletal / Biomechanical | Physical performance | "muscle strength" OR "muscular strength" OR "muscle function" OR "muscular function" OR "muscle power" OR "muscular power" OR "muscle force" OR "muscular force" OR fatigability OR "fatigue resistance" OR "peripheral fatigue" OR "joint moment" OR "joint moments" OR "postural stability" OR posture OR "postural control" OR balance OR sway OR motion OR locomotion OR gait OR walk* OR run* OR jump* OR hop* OR "movement quality" OR "movement pattern" OR "motion pattern" OR coordination OR "motor control" OR "core stability" OR "core strength" OR "trunk stability" OR "trunk strength" OR "lumbopelvic stability" OR "lumbo-pelvic stability" OR "lumbopelvic control" OR "lumbo-pelvic control" | 11 | All Fields |
| | | Anthropo-metrics | Anthropometr* OR "skeletal strength" OR "bone mineral density" OR "bone density" OR "bone mineral content" OR flexib* OR "range of movement" OR "range of motion" | 12 | All Fields |
| | | | #10 OR #11 OR #12 | 13 | |
| | | | #5 AND #9 AND #13 | 14 | |
| | | | Apply human filter | | |

Keywords were divided into main and specific categories for better survey. They were combined using the Boolean operators OR and AND. In order to not mistakenly exclude relevant studies, the Boolean operator NOT was excluded. Medical Subject Headings [Mesh] as a controlled vocabulary thesaurus for indexing and cataloging biomedical literature was applied.

The following databases were screened for publications in English language: PubMed, Web of Science, Embase, Institute of Electrical and Electronics Engineers (IEEE) database as well as ESA's 'Erasmus Experiment Archive', the National Aeronautics and Space Administration's

(NASA) 'Life Science Data Archive' and 'Technical Reports Server' and the German Aerospace Centre's (DLR) database.

## Criteria for included studies

Studies were included if the following inclusion criteria (PICOS) were met:

**Population.** Healthy male and female humans.

**Interventions.** Space flights and ground-based space flight analogues with a duration of ≥ 5 days with a nutritional countermeasure.

**Control.** Space flights and ground-based space flight analogues with a duration of ≥ 5 days without any form of countermeasure.

**Outcomes.** Studies had to contain functional musculoskeletal or cardiopulmonary outcomes.

**Study designs.** Randomised controlled trials (RCT) and controlled clinical trials (CT) were included.

## Data collection and analysis

The Rayyan web application was used to guarantee a blinded screening process throughout [36]. Two independent reviewers performed the initial screening that evaluated a countermeasure (active, passive or nutrition) during space flight, or in a ground-based space flight analogue. Then, two other independent reviewers screened the abstract and title for studies with a nutritional countermeasure (see Fig 1 and S1 Fig). The full text version of the studies was then obtained and screened, and studies were included if the inclusion criteria (PICOS) were met (see Fig 1). In the case of discrepancy between the two main reviewers, a third independent reviewer was employed to resolve any disagreements according to the inclusion criterion.

**Data extraction.** Data were extracted using the AMSRG data Extraction form v2 that is a slightly modified version of that provided by The Cochrane Collaboration (www.cochrane.org) [32]. For all studies containing data presented as means with standard deviations or standard errors, values of all relevant outcome measures for pre and post intervention were extracted in order to calculate the effect sizes. For all studies containing relevant outcome measure data presented as binary outcomes, values for pre and post intervention were extracted and evaluated separately.

Information was extracted to inform study quality assessment via the rating of risk of bias (high, low or unclear) [37]. Method of random sequence generation and allocation concealment was extracted in order to evaluate selection bias [37]. Performance bias was evaluated by information on participants and personnel blinding, with outcome blinding assessment (data processing and analysis) used to evaluate detection bias. Incomplete outcome data, defined attrition bias and evidence of selective reporting, was used to evaluate reporting bias.

**Quality assessment of included studies.** The risk of bias of included studies was assessed by the lead author using the Cochrane Collaboration's (www.cochrane.org) risk of bias analysis tool Version 1 [38] with any uncertainties discussed with the senior author until a consensus was reached. All included studies were scored with "?" representing unclear risk/no information, "+" indicating low risk of bias and "-" representing high risk of bias.

**Quality appraisal of technical principles of included studies.** The quality of the bed rest methodology of the included studies was assessed using the purpose-built AMSRG tool [33] since the search yielded no studies employing any other ground-based analogue. The tool is based on eight criteria detailing how similar a study is to modelling the conditions associated with actual space flight–and thus the 'quality' of its ability to simulate the physiological effects of prolonged μg-exposure. The AMSRG assessment tool ranks the eight criteria from one

 **PRISMA 2009 Flow Diagram**

**Identification**

Records identified through
database searching
(n = 4031)

Additional records identified
through other sources
(n = 0)

**Screening**

Records after duplicates removed
(n = 2695)

Records screened
(n = 2695)

Records excluded
(n = 2674)

**Eligibility**

Full-text articles assessed
for eligibility
(n = 21)

Full-text articles excluded (n = 11)
Wrong study design (n = 7)
Duplicate (n = 1)
Incomplete data presentation (n = 3)

**Included**

Studies included in
qualitative synthesis
(n = 10)

*From:* Moher D, Liberati A, Tetzlaff J, Altman DG, The PRISMA Group (2009). *P*referred *R*eporting *I*tems for *S*ystematic Reviews and *M*eta-*A*nalyses: The PRISMA Statement. PLoS Med 6(7): e1000097. doi:10.1371/journal.pmed1000097

**For more information, visit www.prisma-statement.org.**

**Fig 1. PRISMA flow diagram.** Summary of literature search and screening process. This figure was adapted and modified from Fiebig et al. [35]. CM = countermeasure.

(poor) to eight (excellent) depending on how many of the following criteria are met: (1) Number of bed rest days stated; (2) 6 degrees head down tilt; (3) individualised and controlled diet; (4) set daily routine with fixed wake/sleep time; (5) bed rest phases standardised for all participants; (6) uninterrupted bed rest except for test condition; (7) sunlight exposure prohibited; (8) all measures taken at the same day and time.

**Data analysis.** The quality of the included studies did not allow for a full meta-analysis as the data was too heterogeneous and would break with meta-analysis statistical assumptions. However, in order to provide an overall summary of the data extracted, for all studies containing data presented as means with standard deviations or standard errors, values of all relevant outcome measures effect sizes were calculated by the mean differences between the control and intervention group of pre and post intervention values and bias corrected using the Hedge's G method [39] for small sample sizes, which is a common issue in studies of space flight and ground-based space flight analogues. All calculated effect sizes were defined as small (0.2), medium (0.5), large (0.8) or very large (1.3) as previously described by Rosenthal et al. [40] and were presented in effect size plots with 95% confidence interval error bars. Outcomes were pooled according into the physiological systems they reflect: 'muscle', 'bone' and 'cardiopulmonary'. Muscle outcomes were sub-divided into those that reflect: 'muscle force', 'muscle power', and 'muscle volume'. All data representing a 'positive' effect in favour of the intervention are depicted on the right, and all data representing a 'negative' effect in favour of the control group are presented on the left of the line of no effect (i.e. 0) in effect size plots. For ease of interpretation, all outcomes were presented to show a 'positive' effect as being beneficial, therefore any original outcomes that have negative beneficial effects were inverted by multiplying by -1 for presentation in the overall results (e.g. a drop of resting heart rate results in a negative effect size but it is associated with better general physical fitness and thus is considered a beneficial outcome). All outcomes adjusted by this method are marked with a 'hash' (#). For otherwise qualified studies that did not strictly define a no intervention control group while examining the effect of differential energy intake, the group consuming the lower number of calories was characterized as the control group (vs. the interventional group consuming a greater number of calories). Data that could not be used to calculate Hedge's G values (e.g. binary outcomes) were also extracted and are presented separately. For binary outcomes, finishing the test was considered a positive outcome, therefore more finishers in a results group was interpreted as results favouring that group.

## Results

The initial search leading to definition of the current systematic review performed by Fiebig et al. [35] yielded a total of 4031 studies (Fig 1). After removing duplicates, 2695 studies were left. After screening title and abstract of those 2695 studies, further 2333 studies were excluded, resulting in 278 studies. The 278 studies were then, after another screening with two additional reviewers who screened title and abstracts, divided into three subgroups: passive, active and nutritional countermeasures. The subgroup of nutritional countermeasures for analysis in this systematic review contained a total of 21 studies. After obtaining and reading the full-text versions of these 21 studies, seven studies were excluded because of the wrong study design, one study was excluded due to being a duplicate and three studies were excluded because of incomplete presentation of data, resulting in a total of 10 studies suitable for inclusion.

## Characteristics of included studies

Nine of the included studies were randomised controlled clinical trials (RCT) whilst a single study was a non-randomized controlled clinical trial (CT) (S1 Table). All included studies employed bed rest for durations of between 14 and 60 days as the ground-based analogue to simulate the physiological effects of µg. No actual µg-study met the criteria to be included in this systematic review. A total of 67 participants were included across the studies, 16 of whom were women. The mean number of participants across all studies was 15 (SD = 3) with a mean age of 31 (SD = 6). Six of the included studies incorporated data from the same WISE experimental campaign in Toulouse, France [14,26,41–44]. Two of the ten studies presented binary outcomes in the form of the amount of finishers and non-finishers of an orthostatic tolerance test [25,42]. Across all the studies, 52 outcome parameters were reported consistent with our inclusion strategy (see Table 1 and S1 Table). All studies had a separate control group that did not receive any countermeasure, except for the single CT study that used the same population for intervention and control separated by at least 5 months [25]. Six of the included studies reported muscle, two bone [41,45] and four cardiopulmonary outcomes. Each specific outcome was only reported once, with the exception of number of 'finishers' after an orthostatic tolerance test that was reported by two independent studies employing different methods [25,42].

A total of five different nutritional countermeasures were investigated within the 10 studies. The details of the included nutrition interventions are presented in Table 2.

## Methodological quality of included studies

The overall assessment of risk of bias of the included studies is summarized in Table 3. For the nine RCTs, all domains of bias (selection, performance, detection attrition and reporting bias) included by the Cochrane Risk of Bias Tool were relevant. For the one CT [46], only two domains could be evaluated including attrition bias from incomplete outcome data and reporting bias from selective reporting. One of the nine RCTs had a high risk of selection bias [14], whereas the remaining eight failed to provide sufficient information to be evaluated. All nine RCTs failed to provide sufficient details to evaluate the methods of random sequence generation, allocation concealment, blinding of participants and personnel and blinding of

**Table 2. Nutritional intake.**

| Author + year | Control group | Intervention group |
|---|---|---|
| **Arbeille et al. 2012 Beller et al. 2011 Lee et al. 2014 Scheider et al. 2007 S. Trappe et al. 2008 T.A Trappe et al. 2007** | 1.0 g protein·kg$^{-1}$·day$^{-1}$ | ***Leucine enriched protein diet (LPD)*:** 1.45 g protein·kg$^{-1}$·day$^{-1}$ of dietary protein plus 3.6 g/day of free leucine, 1.8 g/day of free isoleucine, and 1.8 g/day of free valine |
| **Bosutti et al. 2016** | 1.2 g protein·kg$^{-1}$·day$^{-1}$ | ***Protein plus potassium (KHCO$_3$)*:** Control diet plus 0.6 g whey protein·kg$^{-1}$·day$^{-1}$ and 90 mmol KHCO$_3$/day iso-calorically replacing fat and carbohydrate |
| **Florian et al. 2015** | -25% of required daily energy intake (9.0±1.1 MJ/day) | ***Normal energy intake***: Required daily energy intake (9.0±1.1 MJ/day) |
| **Zwart et al. 2005** | Required daily energy intake with protein and carbohydrate accounting for 14% and 59%, respectively. | ***Amino acids plus carbohydrates*:** Control diet plus 16.5 g of essential amino acids and 30 grams of sucrose three times a day |
| **Rejc et al 2015** | 1.2 times resting energy expenditure with 60% of energy as carbohydrate, 25% as fat and 15% as protein. | ***High energy intake*:** Diet containing 1.4 times resting energy expenditure with 60% of energy as carbohydrate, 25% as fat and 15% as protein. |

Daily dietary and supplemental nutrient intake for the control and intervention groups.

**Table 3. Quality assessment of included studies.**

| Author | Random Sequence generation | Allocation concealment | Blinding of participants and personnel | Blinding of outcome assessment | Incomplete outcome data | Selective outcome reporting |
|---|---|---|---|---|---|---|
| Arbeille et al. 2012 | ? | ? | ? | ? | + | - |
| Beller et al. 2011 | ? | ? | ? | ? | + | + |
| Bosutti et al. 2016 | ? | ? | ? | ? | ? | + |
| Florian et al. 2015 | ? | ? | ? | ? | - | + |
| Lee et al. 2014 | - | ? | ? | ? | - | + |
| Rejc et al. 2015 | NA | NA | NA | NA | + | + |
| Schneider et al. 2009 | ? | ? | ? | ? | + | - |
| S. Trappe et al. 2008 | ? | ? | ? | ? | ? | - |
| T.A Trappe et al. 2007 | ? | ? | ? | ? | + | + |
| Zwart et al. 2005 | ? | ? | ? | ? | + | - |

Assessment of risk of bias in the 10 studies included. 'NA' indicates that assessment was not applicable due to study type. '+' indicates low risk, '-' indicates high risk and '?' indicates unclear risk/no information.

outcome assessment. Four of the 10 included studies failed to justify or did not address incomplete outcome data, and four studies failed to report data for all expected outcomes.

## Bed rest methodological quality

The overall assessment of bed rest methodological quality is summarized in Table 4. All included studies employed 6 degree head down tilt bed rest and described its duration, except for a single study that used an angle of 0 degrees in the second phase of the study [46], and one that did not describe the angle of the bed [18]. All studies failed to declare whether exposure to sunlight was prohibited or not. Three of the included studies met all the criteria [14,26,41] described in the AMSRG Bed Rest Assessment Tool v1 [33], except for the prohibition of exposure to sunlight, resulting in a score of seven. One study failed to address or apply all criteria except for bed rest duration, resulting in a score of one [18]. The remaining six studies ranged between three and six.

## Main outcomes parameters

The effect sizes for the physiological systems 'muscle', 'bone' and 'cardiopulmonary' between the intervention and control groups are shown in Figs 2–4, respectively. Overall, the effect of nutritional countermeasure interventions on the investigated outcomes revealed that peak force/cross sectional area (Po/CSA) of myosin heavy chain (MHC) I muscle fibres was 'positive' in favour of the intervention group, supine squat work, calf press work, peak power of MHC IIa fibers, supine squat concentric peak power, peak force of MHC IIa fibers and HR standing were 'negative' in favour of the control group (see PICOS) and 43 outcomes did not show any effect favouring the control or intervention group.

**Effects of nutritional interventions on muscle outcomes.** LPD and protein plus $KHCO_3$ had no clear effect on 26 of the 32 reported muscle outcomes. Supine squat work, supine squat concentric peak power and peak force of MHC IIa muscle fibres showed a large negative effect of LPD. Calf press work and peak power of MHC IIa muscle fibres showed a very large

**Table 4. Bed rest methodological quality.**

| Author | Number of BR days stated | 6° head down tilt | Individualised & controlled diet | Set daily routine with fixed wake/ sleep time | BR phases standardised for all participants | Uninterrupted BR except for test condition | Sunlight exposure prohibited | All measurements taken same day and time | Total score |
|---|---|---|---|---|---|---|---|---|---|
| Arbeille et al. 2012 | Y | Y | N | ? | ? | ? | ? | ? | 2 |
| Beller et al. 2011 | Y | Y | Y | Y | Y | Y | ? | Y | 7 |
| Bosutti et al. 2016 | Y | Y | Y | ? | Y | Y | ? | Y | 6 |
| Florian et al. 2015 | Y | Y | Y | ? | Y | Y | ? | Y | 6 |
| Lee et al 2014 | Y | Y | Y | Y | Y | Y | ? | Y | 7 |
| Rejc et al. 2015 | Y | Y/N | Y | ? | Y | ? | ? | N | 4 |
| Schneider et al. 2009 | Y | Y | Y | Y | Y | Y | ? | Y | 7 |
| S. Trappe et al. 2008 | Y | Y | Y | ? | ? | ? | ? | ? | 3 |
| T.A Trappe et al. 2007 | Y | Y | Y | ? | N | Y | ? | Y | 5 |
| Zwart et al. 2005 | Y | ? | ? | ? | ? | N | ? | N | 1 |

Quality appraised of bed rest method to simulate microgravity to an "ideal design" in the 10 included studies. 'Y' indicates that the criteria was met, 'N' indicates that the criteria was not met and '?' indicates unclear or no information.

negative effect of LPD. Po/CSA of MHC I muscle fibres demonstrated a large positive effect of LPD (Fig 2).

*Muscle volume.* The total of three studies reporting muscle volume found no effect of LPD in the intervention groups [41,43,44].

*Muscle power.* Trappe et al. [44] reported that LPD had a large positive effect on Po/CSA of MHC I muscle fibres, and very large negative effect on peak power of MHC IIa muscle fibres. In contrast, they observed no effect of LPD supplementation on Po/CSA of MHC IIa muscle fibres, normalized power (norm. power) of MHC I and MHC IIa muscle fibres and peak power of MHC I fibres. Trappe et al. [43] had previously reported a large negative effect of LPD on supine squat work and concentric peak power, a very large negative effect on calf press work, but no effect on calf press concentric peak power. Whereas, Rejc et al. [46] found no effect of a hypercaloric diet on maximal explosive power.

*Muscle force.* The four studies investigating muscle force showed no effect of LPD and protein plus KHCO$_3$ on 14 out of 15 outcomes [14,43,44,47]. However, Trappe et al. [44] observed a large negative effect of LPD on peak force of MHC IIa muscle fibres.

**Effects of nutritional interventions on bone outcomes.** Beller et al. [41] found no effect of LPD on bone mineral density (BMD). Likewise, Zwart et al. [18] found no effect of amino acids plus carbohydrates supplementation on bone mineral content (BMC) (Fig 3)

**Effects of nutritional interventions on cardiopulmonary outcomes.** There were no effects of the intervention on VO$_2$max [L/min/kg] and VO$_2$max [L/min] where the effect of protein plus KHCO$_3$ was investigated [47]. However, Schneider et al. [26] found that LPD had a large negative effect on HR when standing, but no effect was observed when supine (Fig 4)

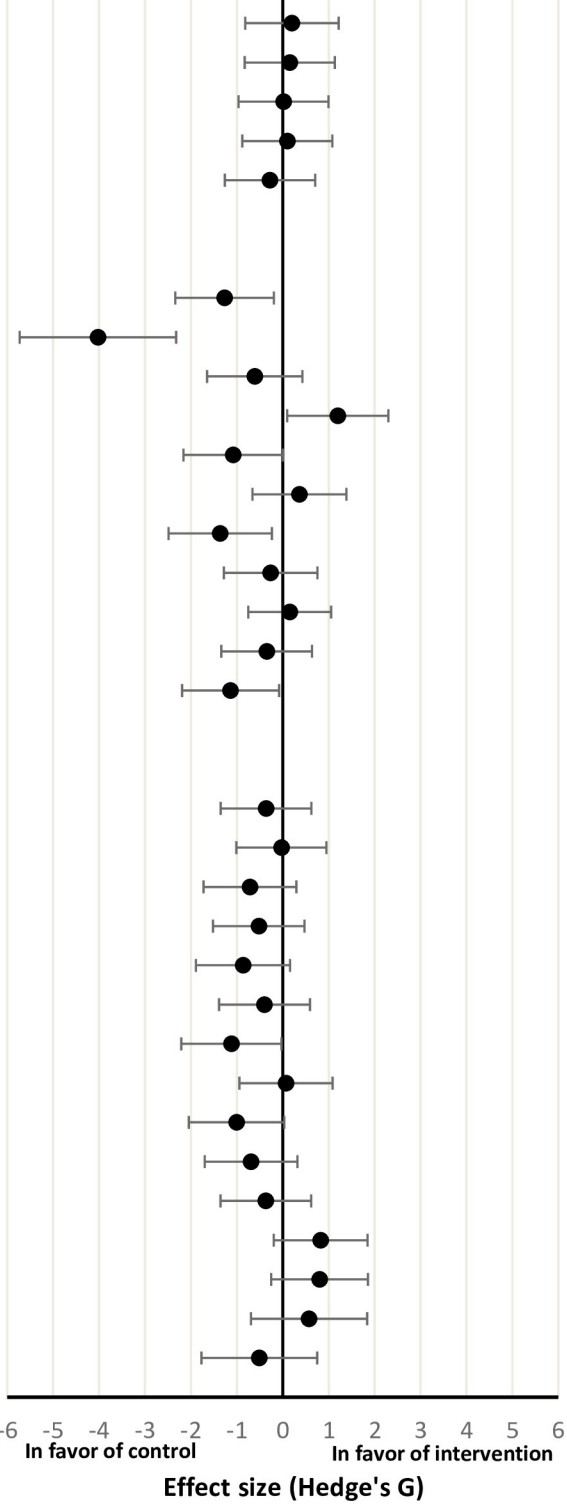

## Muscle volume

S. Trappe 2008   Diameter MHC IIa [μm]   LPD

S. Trappe 2008   Diameter MHC I [μm]   LPD

Beller 2011   Tibia muscle CSA [cm²]   LPD

Beller 2011   Radius muscle CSA [cm²]   LPD

T.A. Trappe  2007   Muscle volume triceps surae [cm³]   LPD

T.A. Trappe 2007   Muscle volume quadriceps femoris [cm³]   LPD

## Muscle power

T.A. Trappe 2007   Supine squat work [J]   LPD

T.A. Trappe  2007   Calf press work [J]   LPD

S. Trappe 2008   Po/CSA MHC IIa [kN/m³]   LPD

S. Trappe 2008   Po/CSA MHC I [kN/m³]   LPD

S. Trappe 2008   Norm. power MHC IIa [W/L]   LPD

S. Trappe 2008   Norm. power MHC I [W/L]   LPD

S. Trappe 2008   Peak power MHC IIa [μN · FL⁻¹ · s⁻¹]   LPD

S. Trappe 2008   Peak power MHC I [μN · FL⁻¹ · s⁻¹]   LPD

Rejc 2014   Max explosive power [W]   High energy intake

T.A. Trappe 2007   Calf press concentric peak power [W]   LPD

T.A. Trappe 2007   Supine squat concentric peak power [W]   LPD

## Muscle force

T.A. Trappe  2007   Calf press eccentric peak force [N]   LPD

T.A. Trappe 2007   Calf press concentric peak force [N]   LPD

T.A. Trappe 2007   Calf press isometric force [N]   LPD

T.A. Trappe 2007   Supine squat eccentric peak force [N]   LPD

T.A. Trappe 2007   Supine squat concentric peak force [N]   LPD

T.A. Trappe 2007   Supine squat isometric force [N]   LPD

S. Trappe 2008   Peak force MHC IIa [mN]   LPD

S. Trappe 2008   Peak force MHC I [mN]   LPD

Lee 2014   Leg strenght press [Kg]  LPD

Lee 2014   Knee extension total work [Nm]   LPD

Lee 2014   Dorsi flexion peak torque [Nm]   LPD

Lee 2014   Plantar flexion peak torque [Nm]   LPD

Lee 2014   Knee extension peak torque [Nm]   LPD

Bosutti 2016   Max plantar flexion torque [Nm]   Protein + KHCO₃

Bosutti 2016   Max knee extension torque [Nm]   Protein + KHCO₃

-6  -5  -4  -3  -2  -1  0  1  2  3  4  5  6

**In favor of control**          **In favor of intervention**

**Effect size (Hedge's G)**

**Fig 2.  Effect size plot of operationally relevant muscle outcomes.** Effect size plot of operationally relevant muscle outcomes categorized into 'muscle volume', 'muscle power' and 'muscle force'. Effect sizes were calculated by the mean differences between the control and intervention group of pre and post bed rest values with Hedges' G and bias corrected for sample size with a confidence interval of 95%. All calculated effect sizes were defined as small

(0.2), medium (0.5), large (0.8) or very large (1.3). The right direction on the x-axis indicates a positive effect of the intervention and the left direction on the x-axis indicates a negative effect of the intervention. CSA = cross sectional area; LPD = leucine protein diet; max = maximum; MHC = myosin heavy chain; Po = peak force.

**Effects of nutritional interventions on orthostatic tolerance.** Two of the included studies investigated orthostatic tolerance after bed rest with Florian et al. [25] employing a lower body negative pressure (LBNP) test applying a pressure of -15, -30 and -45 mmHg for 7 minutes each, or until presyncope, and Arbeille et al. [42] inducing a 80 degree tilt for 10 minutes that was then supplemented by LBNP exposure that increased by -10 mmHg every 3 minutes until -50 mmHg (Table 5). Both studies had one more finisher in the intervention group, i.e. five in the hypercaloric [25], and six in the LPD intervention groups [42] out of eight participants, compared to the control groups that had four and five finishers, respectively.

## Discussion

The main findings of this study were: (1) substantial heterogeneity of reported outcomes across studies, (2) the inconsistent quality of methods used to minimize risk of bias and

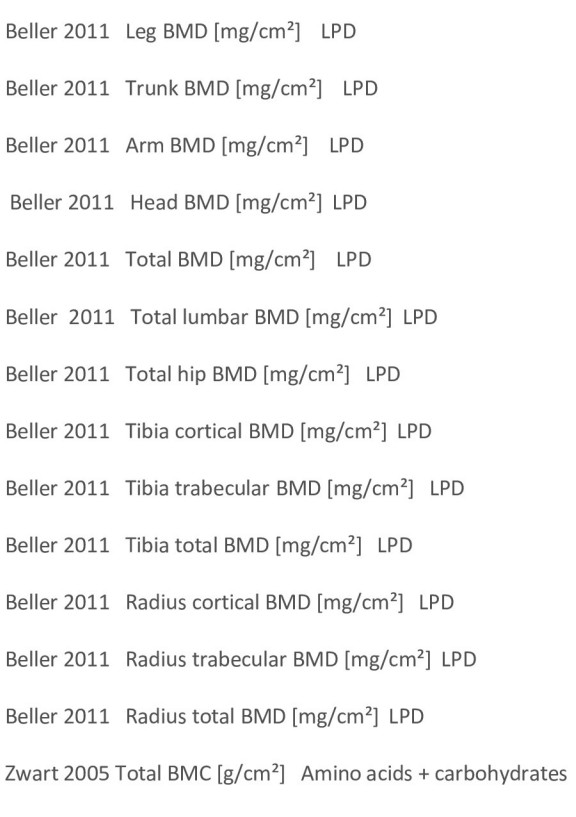
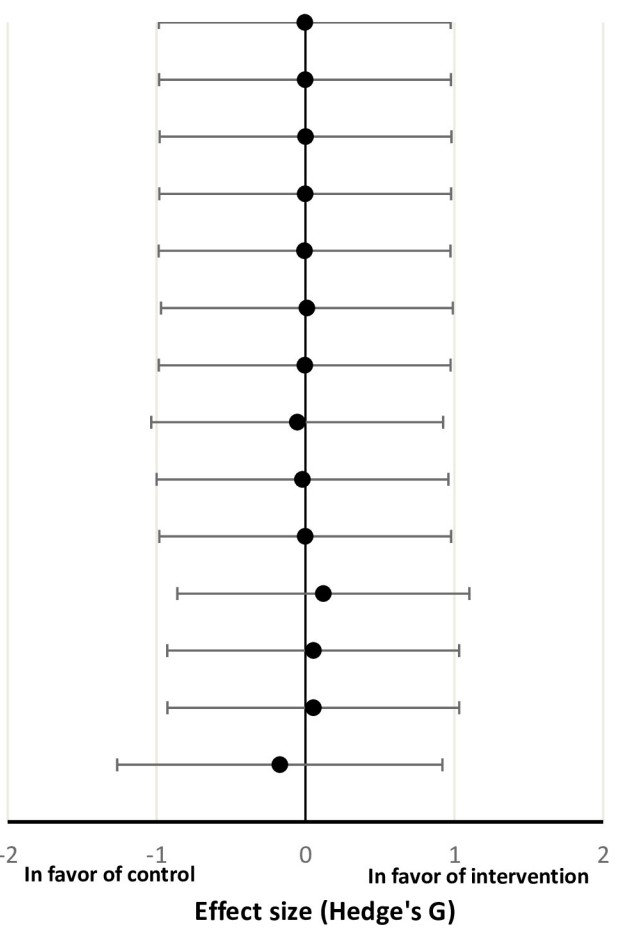

**Fig 3. Effect size plot of operationally relevant bone outcomes.** Effect sizes were calculated by the mean differences between the control and intervention group of pre and post bed rest values with Hedges' G and bias corrected for sample size with a confidence interval of 95%. All calculated effect sizes were defined as small (0.2), medium (0.5), large (0.8) or very large (1.3) The right direction on the x-axis indicates a positive effect of the intervention and the left direction on the x-axis indicates a negative effect of the intervention. BMC = bone mineral content; BMD = bone mineral density; LPD = leucine protein diet.

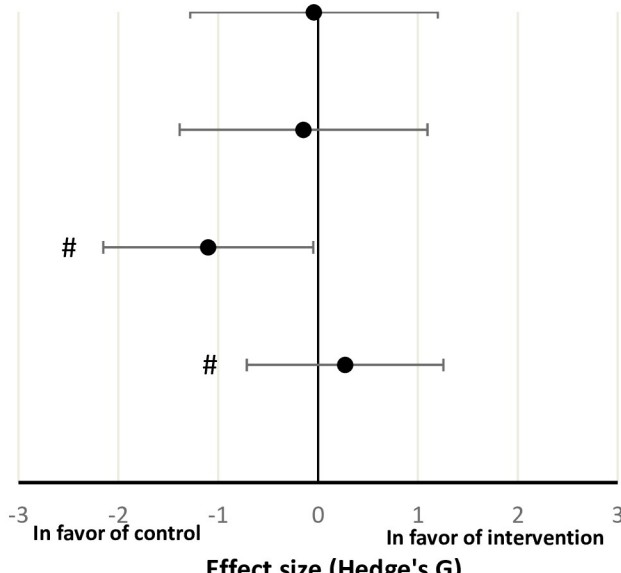

**Fig 4. Effect size plot of operationally relevant cardiopulmonary outcomes.** Effect sizes were calculated by the mean differences between the control and intervention group of pre and post bed rest values with Hedges' G and bias corrected for sample size with a confidence interval of 95%. All calculated effect sizes were defined as small (0.2), medium (0.5), large (0.8) or very large (1.3). The right direction on the x-axis indicates a positive effect of the intervention and the left direction on the x-axis indicates a negative effect of the intervention. HR = heart rate; LPD = leucine protein diet; VO2max = volume oxygen maximum.

methodology of bed rest in the included studies, and (3) an absence of evidence for meaningful effects in favour of standalone nutritional countermeasure interventions on musculoskeletal and cardiopulmonary outcomes during 14 to 60 days of bed-rest ground-based analogues, with a tendency towards detrimental effects on specific muscle outcomes associated with power in the lower extremities.

The present data demonstrate an absence of meaningful effects when nutritional counter-measures were used as a standalone intervention during bed rest to ameliorate the adverse effects of μg on the musculoskeletal and cardiopulmonary systems. Moreover, this systematic review was unable to find eligible in flight-data from astronauts or any other ground-based space flight analogues than bed rest for inclusion that met the pre-specified inclusion criteria (PICOS). As a result, only bed rest studies were included. Six of the 10 included studies reported data from the WISE 2005 campaign investigating the effects of 60 days of 6 degrees head down bed rest in 24 women of which only the 16 women assigned to the nutrition and control groups were evaluated in this systematic review. The WISE 2005 campaign was

**Table 5. Binary outcome data.**

| Author + year | BR days | Population | Study Design | Intervention | Outcome measures | Applied test/ | Control group | Intervention group |
|---|---|---|---|---|---|---|---|---|
| **Arbeille et al. 2012** | 60 | 16 | RCT | LPD | Number of finishers | Tilt + LBNP test after bed rest | 5 out of 8 finishers | 6 out of 8 finishers |
| **Florian et al. 2015** | 14 | 9 | RCT | Normal energy intake | Number of finishers | LBNP test after bed rest | 4 out of 8 finishers | 5 out of 8 finishers |

Binary outcome data for orthostatic tolerance that could not be included in the effect size plots presenting the number of finishers of tilt + lower body negative pressure test (LBNP) or standalone LBNP test. BR = bed rest; RCT = randomized controlled trial.

investigating how the physiology of women might adapt to space flight and to test the ability of nutrition as a standalone countermeasure and exercise against the expected physiological adaptations associated with bed rest. The remaining four studies analysed in this systematic review were highly heterogeneous in terms of sample sizes, gender composition, the nature of the nutritional countermeasures tested, and the duration of bed rest. The investigated outcomes across all included studies (including the WISE 2005 studies) were highly heterogeneous with no specific outcomes tested more than once, with the exception of orthostatic tolerance that was investigated by two independent studies using slightly different methods [25,42].

## Quality of evidence and overall completeness

The present systematic review sought to investigate physiological outcomes in the domains of musculoskeletal and cardiopulmonary systems with a demonstrable space flight operational relevance (as defined by operational experts of ESA's Space Medicine Team). However, only a small proportion of the pre-defined operationally relevant outcomes (See Table 1 and S1 Table) were presented in the included studies. Moreover, the reporting of outcomes of the included studies was rather poor. Every included study reported a unique set of physiological outcomes. The absence of a standard data set across studies prevented data pooling and meta-analysis, therefore current findings and operational space medicine recommendations are based on single studies that could be flawed or biased, rather than a number of repeated and standardised studies. The overall statistical power of the findings is, therefore, limited and again could be improved by conducting repeated homogenous studies that would increase the overall total sample size when pooled.

**Bed rest methodology.** Bed rest studies seek to simulate μg and its impact on human physiology as close as possible to the actual μg environment in space and it has been concluded that head down bed is a valid simulation model for most physiological effects of spaceflight [48]. Bed rest is an established terrestrial analogue for axial unloading. Compared to actual space flight, bed rest studies have the advantage that they have fewer confounding factors, and bed rest provides the most common method to investigate the effectiveness of nutritional spaceflight countermeasures [48]. The bed rest methodology of included studies was good with most studies matching the "ideal design" of a bed rest study [49]. However, all studies failed to address whether sunlight exposure was prohibited which is problematic as sunlight exposure stimulates vitamin D synthesis that might itself act as a countermeasure [50], and potentially affect the physiological systems investigated in this systematic review. However, vitamin D supplementation has been tested in actual space flight and it was found that it was inefficient to ameliorate bone loss in space [9]. In addition, seven out of 10 studies did not report set daily routine with fixed wake/sleep time for all participant which might impact musculoskeletal and cardiovascular outcomes as sleep affects muscle strength [51] and athletic performance [52]. Overall most aspects of the assumed "ideal design" of bed rest such as stating the number of days for bed rest and the use of 6 degree head down tilt were met and the data can be considered transferable to actual astronauts in the μg environment of space. However, despite the high methodological quality of the included studies, bed rest is not a perfect method to simulate the actual space flight environment as it does not eliminate the influence of gravity and fails to remove the loading G-vector from the chest to the back [48]. Therefore, it is important to exercise caution when interpreting results exclusively based on bed rest studies and applying them to actual space flight situations, as it has been suggested that bed rest might be a less reliable analogue for important physiological parameters such as spinal

dysfunction and fluids shifts which might affect the cardiopulmonary outcomes such as orthostatic tolerance and HR investigated in the current systematic review [48].

**Risk of bias.** Overall, the risk of bias of the included studies was high, with seven of the studies reporting incomplete outcome data and/or reporting selective outcomes. Eight of the nine included RCTs failed to sufficiently describe the randomization of the participants [25,26,41–45,47] and Lee et al. [14] described a non-random component in the sequence generation. In addition, all included studies failed to sufficiently describe methods for allocation concealment, blinding of participants and personnel, and blinding of outcome assessment. The overall high risk of bias of the included studies highlights the need for a better standardization of bed rest studies investigating nutritional countermeasures. As a result of the high risk of bias, as well as the heterogeneous study designs and outcomes of the included studies, findings of the present systematic review have to be interpreted with caution. The overall quality of evidence of the included studies could be improved by providing more methodological information addressing the Cochrane criteria for risk of bias [53].

## For effects of nutritional interventions on muscle outcomes

In the current systematic review, LPD was the only nutritional countermeasure intervention leading to meaningful effects on any of the investigated outcomes across the investigated physiological systems. The effects of LPD on muscle parameters were limited to only one outcome (Po/CSA MHC) showing a significant positive effect [54]. In contrast, supine squat work, supine squat concentric peak power, peak power MHC IIa, peak force MHC IIa and calf press work showed a negative effect of LPD. The remaining 26 outcomes showed no effect of any of investigated nutritional countermeasure interventions at all. Interestingly, five of the six outcomes demonstrating an effect of LPD were in the group of 11 investigated muscle power outcomes, two of which showed a large negative effect and two which showed a very large negative effect, whereas only one outcome showed a large positive effect of LPD. This suggests that LPD had a negative effect on four out of the 11 specific muscle outcomes associated with power investigated in this systematic review. However, all these data are derived from the same participants in the 2005 WISE campaign [43,44] and, therefore, these results must be interpreted cautiously. Overall, protein and amino acid supplementation are the most extensively investigated nutritional countermeasure (LPD being evaluated in 45 of the 52 outcomes in this study) against muscular deconditioning in unloaded humans [9,55]. The efficacy of protein and amino acid supplementation to mitigate the physiological effects on muscle integrity and function caused by μg-exposure are inconclusive, with studies showing diverging results with no clear effects [43,44,55]. Furthermore, some studies suggested that protein supplementation has no additional protective effect upon muscle mass and function when energy and protein consumption are adequate [56,57]. The findings of the current systematic review go beyond this, suggesting that leucine might have a negative effect on power and force muscle outcomes. In fact, a recent critical review reinforced this, as the authors concluded that leucine as a standalone countermeasure intervention is ineffective against muscle loss [58]. Interestingly, however, a novel amino acid composition has shown the ability to preserve muscle mass during muscle disuse-induced atrophy by unilateral knee immobilization in young men (43) but has yet to be tested in actual space flight or space flight analogues. The current systematic review does not, however, provide any support for amino acids (LPD) having a positive effect on muscle health during unloading. In conclusion, this systematic review does not support a beneficial effect of the tested standalone nutritional interventions to protect muscles from deconditioning in simulated μg, in fact there is even a tendency towards detrimental effects on specific muscle outcomes associated with power in the lower extremities.

### Effects of nutritional interventions on bone outcomes

None of the 14 investigated outcomes showed an effect of standalone nutritional countermeasures. However, the relatively weak effect of bed rest that was observed in the two investigated studies on BMD and BMC [18,41] limits the sensitivity to any potential amelioration of nutritional countermeasure interventions [18,41]. Hence, no conclusions of the effect of LPD or amino acids plus carbohydrates on bone can be drawn from the included studies. In the scientific literature the effect of nutritional countermeasures on bone have revealed diverging results depending on the population and concurrent dietary intake of other nutrients [9,17]. Negative effects of protein supplementation observed on bone have been suggested to result from sulphur-containing amino acids increasing bone resorption and/or induction of metabolic acidosis depleting skeletal bicarbonate stores, and thus bone mass [9]. Therefore, it has been proposed that high protein diets containing sulphuric acids should be provided with base precursors to counteract the adverse effects [18]. However, the current systematic review did not report any data supporting any protective effect of standalone nutritional interventions on bone properties in simulated µg.

### Effects of nutritional interventions on cardiopulmonary outcomes

In the study by Schneider et al. [26], LPD had a large negative effect on HR when standing, but no effect when measured in supine position. Bosutti et al. [47] found no effect of the intervention when investigating protein plus $KHCO_3$ on $VO_2$max [L/min/kg] and $VO_2$max [L/min]. It has been proposed that protein might be able to mitigate the decline in $VO_2$peak observed after space flight by mitigating the concurrent cardiac muscle atrophy observed after spaceflight and bed rest [26,59]. However, this finding is not supported in the results of this systematic review. As such, the available data do not support any protective effect of standalone nutritional interventions on cardiovascular properties in simulated µg.

Overall, this systematic review found little evidence supporting the effect of any standalone nutritional countermeasures to mitigate physiological adaptations on operationally relevant musculoskeletal and cardiopulmonary parameters during exposure to µg.

## Conclusions

The main findings of this study were: (1) the heterogeneity of reported outcomes across studies, (2) the inconsistency of the methodology of the included studies, and (3) an absence of meaningful effects of standalone nutritional countermeasure interventions on musculoskeletal and cardiovascular outcomes, with a tendency towards detrimental effects on specific muscle outcomes associated with power in the lower extremities. This, however, does not mean that nutritional countermeasures may have no effect on operationally relevant outcomes at all as nutritional countermeasures may affect the musculoskeletal and cardiopulmonary systems differently when combined with other countermeasures such as resistance exercise. This systematic review, rather, highlights the paucity of data investigating the effect of nutrition as a standalone countermeasure on operationally relevant outcome parameters of the musculoskeletal and cardiopulmonary systems. Based on the evidence of the included studies, it cannot be expected that nutrition alone will be effective in maintaining musculoskeletal and cardiopulmonary integrity in response to gravitational unloading.

## Supporting information

**S1 Fig. Summary of overall literature search and screening process.** Initial literature search and screening procedure performed by Leonie et al. 2018. Literature screening was performed

using Rayyan web application. CM = countermeasure. (Figure adapted from Fiebig et al. [35]).
(TIF)

**S1 Table. Study characteristics.**
(DOCX)

**S2 Table. PRISMA 2009 checklist.**
(DOCX)

**S1 File. Review protocol.**
(DOCX)

## Acknowledgments

This systematic review was performed as part of a project by the Aerospace Medicine Systematic Review Group and the Space Medicine Team of the European Space Agency at the European Astronaut Centre (EAC). Special thanks are given to Dr Anna Fogtman and Robert Ekman from the EAC and Dr Bernd Johannes from the German Aerospace Center for their support during the project.

## Author Contributions

**Conceptualization:** Leonie Fiebig, Tobias Weber.

**Data curation:** Peter H. Sandal, David A. Green, Tobias Weber.

**Formal analysis:** Peter H. Sandal, David A. Green, Tobias Weber.

**Funding acquisition:** Tobias Weber.

**Investigation:** Peter H. Sandal, Tobias Weber.

**Methodology:** Peter H. Sandal, Leonie Fiebig, David A. Green, Tobias Weber.

**Project administration:** Tobias Weber.

**Resources:** Tobias Weber.

**Software:** Peter H. Sandal, Tobias Weber.

**Supervision:** Andrew Winnard, Nick Caplan, David A. Green, Tobias Weber.

**Validation:** Tobias Weber.

**Visualization:** Tobias Weber.

**Writing – original draft:** Peter H. Sandal.

**Writing – review & editing:** David Kim, Leonie Fiebig, Andrew Winnard, Nick Caplan, David A. Green, Tobias Weber.

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
