## [Decision Letter · Decision Letter 0]

15 May 2020

PONE-D-20-10013

Effectiveness of nutritional countermeasures in microgravity and its ground-based analogues to ameliorate musculoskeletal and cardiopulmonary deconditioning – A Systematic Review

PLOS ONE

Dear Dr Weber,

Thank you for submitting your manuscript to PLOS ONE. After careful consideration, we feel that it has merit but does not fully meet PLOS ONE’s publication criteria as it currently stands. Therefore, we invite you to submit a revised version of the manuscript that addresses the points raised by the reviewer and editorial comments during the review process.

Though the authors have conducted an extensive survey of nutritional countermeasures to  musculoskeletal and cardiopulmonary challenges in microgravity conditions, I suggest improving  the quality of all figures, details in all legends including the  supplemental information provided.

We would appreciate receiving your revised manuscript by Jun 29 2020 11:59PM. To enhance the reproducibility of your results, we recommend that if applicable you deposit your laboratory protocols in protocols.io, where a protocol can be assigned its own identifier (DOI) such that it can be cited independently in the future. For instructions see: http://journals.plos.org/plosone/s/submission-guidelines#loc-laboratory-protocols

We look forward to receiving your revised manuscript.

Kind regards,

Dr. Sakamuri V. Reddy

Academic Editor

PLOS ONE

Additional Editor Comments:

In this manuscript, Dr. Weber and colleagues performed a systematic review to evaluate the effectiveness of nutritional countermeasure to ameliorate the physiological challenges of the musculoskeletal and cardiopulmonary systems in microgravity conditions necessary to maintain human health during space flight missions. The survey concluded that nutrition alone cannot be expected to be an effective countermeasure in maintaining musculoskeletal and cardiopulmonary integrity during space flight and bed rest. The authors have performed systematic review with a thorough background literature. Specific comments to further improve the manuscript are as follow.

Title- The authors may consider rephrasing the title for a review article- “Nutritional countermeasures to musculoskeletal and cardiopulmonary challenges in microgravity conditions”.

They have conducted a Systematic Review and Meta-Analyses of data. They have used PRISMA 2009 flow diagram. I suggest the authors to verify the journal policy and using statistical methods appropriately. Also, improve the quality of all figures provided. Fig.1 legend, please provide ref#35 for Fiebig et al 2018 noted.

References: Please remove the repeated information for references provided ex., Ref#21: “Available from:----------” verify complete details for Ref#37 “AMSRG Data extraction form v2. 2017” following a journal article for citations listed.

Although the Methods section include Data analysis, I suggest providing details in legends with for Figures 1-4 with statistical methods and significance data (p.15); also, details in legends for “Supporting information” data provided.

2. Please provide the date range for when the literature was searched.

'The author(s) received no specific funding for this work.'

We note that one or more of the authors are employed by a commercial company: KBR GmbH

Reviewers' comments:

Reviewer's Responses to Questions

**Comments to the Author**

1. Is the manuscript technically sound, and do the data support the conclusions?

Reviewer #1: Yes

2. Has the statistical analysis been performed appropriately and rigorously? 

Reviewer #1: Yes

3. Have the authors made all data underlying the findings in their manuscript fully available?

Reviewer #1: Yes

4. Is the manuscript presented in an intelligible fashion and written in standard English?

Reviewer #1: Yes

5. Review Comments to the Author

Reviewer #1: This is a very elaborate and detailed review regarding the effect of microgravity on the health of Astronauts during prolonged space flight. The authors have analyzed with effective statistical tools published data from several papers related to this field and have reported the results of their analysis on bone density, cardiovascular health, renal, vitamins, minerals, orthostatic hypotension,immune system and other relevant physiological changes pertaining to disturbance of normal homeostasis during prolonged space flight.This is a very well researched manuscript and could be a great source for information for biologist and physiologists doing microgravity research.

6. PLOS authors have the option to publish the peer review history of their article (what does this mean?). If published, this will include your full peer review and any attached files.

Reviewer #1: Yes: Hirendra Nath Banerjee

---

## [Author Response · Author response to Decision Letter 0]

22 May 2020

Comments from Dr Sakamuri V. Reddy:

Though the authors have conducted an extensive survey of nutritional countermeasures to musculoskeletal and cardiopulmonary challenges in microgravity conditions, I suggest improving the quality of all figures, details in all legends including the supplemental information provided.

Thank you. We have done as suggested.

Comments from Dr Prof Banerjee:

Dear Professor Banerjee,

Thank you for your helpful suggestions and comments. Please find your comments highlighted in red color, followed by our answers in black color below. All changes in the revised manuscript have been highlighted using the track changes function in word.

In this manuscript, Dr Weber and colleagues performed a systematic review to evaluate the effectiveness of nutritional countermeasure to ameliorate the physiological challenges of the musculoskeletal and cardiopulmonary systems in microgravity conditions necessary to maintain human health during space flight missions. The survey concluded that nutrition alone cannot be expected to be an effective countermeasure in maintaining musculoskeletal and cardiopulmonary integrity during space flight and bed rest. The authors have performed systematic review with a thorough background literature. Specific comments to further improve the manuscript are as follow.

Thank you for your thorough review!

Title- The authors may consider rephrasing the title for a review article- “Nutritional countermeasures to musculoskeletal and cardiopulmonary challenges in microgravity conditions”.

Thanks for proposing a new title. However, the authors of the present manuscript feel that the original title better reflects study design and purpose of the study, so we decided to keep the title as it is.

They have conducted a Systematic Review and Meta-Analyses of data. They have used PRISMA 2009 flow diagram. I suggest the authors to verify the journal policy and using statistical methods appropriately. 

Thank you for highlighting this, and apologies if this was not clear. We have not done a full meta-analysis due to the high heterogeneity of study designs and outcomes. More precisely, we did not do a meta-analysis, as either the RCT/CT data was too heterogeneous, or there were only within participant designs which would break the meta analysis statistical assumptions. However, to still provide an overall summary, we converted all included data to effect sizes in standardised units to give an overall comparable summary of the evidence base as it stands today. In conclusion, we have used statistical methods appropriately and the present methodology is in line with the highest standards available for systematic literature reviews (Cochrane guidelines). We have added a sentence to the data analysis section to make this clear. 

Also, improve the quality of all figures provided. Fig.1 legend, please provide ref#35 for Fiebig et al 2018 noted.

Thank you. We have done as suggested. 

References: Please remove the repeated information for references provided ex., Ref#21: “Available from:----------” 

Thank you. We have done as suggested. 

Verify complete details for Ref#37 “AMSRG Data extraction form v2. 2017” following a journal article for citations listed.

Thank you. We have verified this and revised the references in the bibliography. 

Although the Methods section include Data analysis, I suggest providing details in legends with for Figures 1-4 with statistical methods and significance data (p.15); also, details in legends for supporting information data provided.

Thank you. We have done as suggested

This is a very elaborate and detailed review regarding the effect of microgravity on the health of Astronauts during prolonged space flight. The authors have analyzed with effective statistical tools published data from several papers related to this field and have reported the results of their analysis on bone density, cardiovascular health, renal, vitamins, minerals, orthostatic hypotension,immune system and other relevant physiological changes pertaining to disturbance of normal homeostasis during prolonged space flight. This is a very well researched manuscript and could be a great source for information for biologist and physiologists doing microgravity research.

Once again, thank you for your thorough review!

Thank you. We verified this and can confirm that the PLOSONE requirements are met. 

2. Please provide the date range for when the literature was searched.

Thank you. We have done as suggested and added this information to the methods section.

'The author(s) received no specific funding for this work.'

We note that one or more of the authors are employed by a commercial company: KBR GmbH

Thank you. We have done as suggested and added this information to the author contribution section.

Not applicable.

Not applicable.

We have done as suggested and added this information to the author contribution section.

We have done as suggested.

This has been noted, and the PLOS ONE information on competing interests has been reviewed, and the corresponding author has declared all potential conflicts of interest on behalf of all contributing authors.

We have done as suggested.

---

## [Editor Report · Decision Letter 1]

27 May 2020

Effectiveness of nutritional countermeasures in microgravity and its ground-based analogues to ameliorate musculoskeletal and cardiopulmonary deconditioning – A Systematic Review

PONE-D-20-10013R1

Dear Dr. Weber,

We are pleased to inform you that your manuscript has been judged scientifically suitable for publication and will be formally accepted for publication once it complies with all outstanding technical requirements.

With kind regards,

Dr. Sakamuri V. Reddy

Academic Editor

PLOS ONE
---

## [Editor Report · Acceptance letter]

29 May 2020

PONE-D-20-10013R1 

Effectiveness of nutritional countermeasures in microgravity and its ground-based analogues to ameliorate musculoskeletal and cardiopulmonary deconditioning – A Systematic Review 

Dear Dr. Weber:

I am pleased to inform you that your manuscript has been deemed suitable for publication in PLOS ONE. Congratulations! Your manuscript is now with our production department. 

With kind regards,

on behalf of

Dr. Sakamuri V. Reddy 

Academic Editor

PLOS ONE